# High-Throughput Computational Search for Half-Metallic Oxides

**DOI:** 10.3390/molecules25092010

**Published:** 2020-04-25

**Authors:** Laalitha S. I. Liyanage, Jagoda Sławińska, Priya Gopal, Stefano Curtarolo, Marco Fornari, Marco Buongiorno Nardelli

**Affiliations:** 1Department of Physics, University of North Texas, Denton, TX 76203, USA; jagoda.slawinska@gmail.com (J.S.); priyagviji@gmail.com (P.G.); 2Faculty of Computing and Technology, University of Kelaniya, Kelaniya 11600, Sri Lanka; 3Center for Autonomous Materials Design, Duke University, Durham, NC 27708, USA; stefano@duke.edu; 4Materials Science, Electrical Engineering, Physics and Chemistry, Duke University, Durham, NC 27708, USA; 5Department of Physics, Central Michigan University, Mount Pleasant, MI 48859, USA; forna1m@cmich.edu

**Keywords:** half metals, transition metal oxides, high-throughput search, aflowlib, spintronics

## Abstract

Half metals are a peculiar class of ferromagnets that have a metallic density of states at the Fermi level in one spin channel and simultaneous semiconducting or insulating properties in the opposite one. Even though they are very desirable for spintronics applications, identification of robust half-metallic materials is by no means an easy task. Because their unusual electronic structures emerge from subtleties in the hybridization of the orbitals, there is no simple rule which permits to select a priori suitable candidate materials. Here, we have conducted a high-throughput computational search for half-metallic compounds. The analysis of calculated electronic properties of thousands of materials from the inorganic crystal structure database allowed us to identify potential half metals. Remarkably, we have found over two-hundred strong half-metallic oxides; several of them have never been reported before. Considering the fact that oxides represent an important class of prospective spintronics materials, we have discussed them in further detail. In particular, they have been classified in different families based on the number of elements, structural formula, and distribution of density of states in the spin channels. We are convinced that such a framework can help to design rules for the exploration of a vaster chemical space and enable the discovery of novel half-metallic oxides with properties on demand.

## 1. Introduction

Spintronics attempts to employ electron’s charge and spin degrees of freedom in novel computing and data storage applications, assumed to be faster and more energy efficient than their conventional counterparts [1,2]. Successful development of such devices strongly depends on the availability and integration of diverse materials which would enable harnessing of electrons spins. Among several obstacles on the route to the practical realization of both spin logic and memory elements is the lack of sufficiently spin-polarized magnetic electrodes that could act as spin current sources. The mixed spin current may not only impede the efficient spin injection, but also limit the performance of devices utilizing either giant (GMR) or tunneling magnetoresistance (TMR) [3]. In this regard, half-metals (HM), which are fully spin-polarized at the Fermi level [4,5,6,7] and only pass a spin-up or spin-down current, emerge as natural candidates to use as electrodes in such devices. The identification of half-metallic compounds integrable with mature architectures based on complementary metal oxide semiconductors (CMOS) would therefore represent an important step towards broader implementation of spintronics. Regrettably, half metals are extremely rare in nature, and discovery of materials with suitable properties is challenging, either from experimental or theoretical side [8,9,10,11].

Most representative half-metallic compounds belong to either heusler/half-heusler alloys (e.g., NiMnSb and PtMnSb [12]) or transition metal oxides (TMO) in multiple structural forms, such as simple rutiles (CrO${}_{2}$ [13]), spinels (Fe${}_{3}$O${}_{4}$ [14]), perovskites (La${}_{1-x}$Sr${}_{x}$MnO${}_{3}$ [15]), pyrochlores (Tl${}_{2}$Mn${}_{2}$O${}_{7}$ [16]), or double perovskites (Sr${}_{2}$FeMoO${}_{6}$ [17]). Interestingly, extensive studies of these and similar materials have clearly shown that half-metallicity, despite being quite peculiar, does not manifest in an obvious way in magnetic, electrical, or optical properties that could be measured. The simplest indicator is perhaps the integer magnetic moment per unit cell at absolute zero, which is a necessary condition for half-metallicity in a stoichiometric compound. However, it does not permit unambiguously distinguishing a half metal from a standard ferromagnet. Observation of conducting electrons of one spin direction could certainly provide a more direct proof, but experimental techniques such as spin-polarized photoemission or transport measurements in point contacts and tunnel junctions are flawed by uncertainty and full spin polarization is rarely confirmed. Electronic structure calculations are thus the most useful tool to identify half metals, even though the existence of a band gap for just one spin channel is by some means fortuitous and hard to predict. In fact, the design rules so far have strongly relied on investigations in specific crystal families based on a prototype.

Here, we have identified a large number of new candidates for half metals by performing a high-throughput (HT) screening of the electronic structure information from the aflowlib database [18,19,20]. Currently, the repository contains over 3,000,000 different materials entries, among which 60,324 belong to the inorganic crystal structure database (ICSD) [21], representing fully determined and synthesized compounds. The data from this subset have been explored based on the analysis of their calculated density of states (DOS); the presence of a band gap in just one spin channel has been adapted as a major criterion for half-metallicity. The search revealed in total over one-thousand potential half metals, including most of the known examples. We have selected and described in detail a subgroup of oxides, among which 223 materials have been predicted to be strongly half-metallic. We have further classified them according to the number of elements and atomic structure, recognizing new crystal prototypes. Remarkably, the reported electronic structure parameters have shed more light on the mechanisms of hybridization and origin of half-metallicity. Figure 1 shows the schematic diagram of the full search procedure as well as the classification of the identified compounds.

## 2. Results

Before we start with an overview of the HT search, let us remark that materials repositories usually do not provide complete information on the magnetic ordering. Thus, some ferromagnetic compounds may be classified as paramagnetic and they cannot be included in this search. Conversely, several materials labeled as ferromagnetic may actually display different magnetic orders. As the ground state needs to be verified experimentally or theoretically via more realistic ab initio calculations, we will hereafter refer to potential half metals. In the screening procedure, we have first selected materials whose spin polarization at the Fermi level (E${}_{F}$) is different from zero. This can be expressed as P${}_{0}$(E${}_{F}$) = [N${}_{↑}-$ N${}_{↓}$)]/[N${}_{↑}$ + N${}_{↓}$] ≠ 0, where N${}_{↑/↓}$ denotes the density of states for spin majority/minority at E${}_{F}$. Second, we have analyzed electronic states around the Fermi level, and classified the compounds as metals, half-metals, and semiconductors/insulators. We have further restricted the analysis to “half-metallic semiconductors” determined by conditions on the band gap (E${}_{g}$ < 3.5 eV), valence band maximum (VBM > −1.5 eV), and conduction band minimum (CBM < 2.5 eV) of the insulating channel. Finally, in order to unveil materials that are strongly half-metallic, we have put a constraint on the spin polarization in the energy region limited by VBM and CBM, namely, P${}_{0}$(E${}_{\mathrm{gap}}$) > 0.8. The spin-flip excitation energies of the transition from majority to minority spin, and vice versa, have been also constrained to ensure robustness of half-metallicity with respect to thermal fluctuations [4].

The high-throughput procedure revealed a total number of 223 candidates for strongly half-metallic oxides. They have been categorized according to the number of elements; we have classified 13 binary, 105 ternary, 101 quaternary, and four quinary compounds. Each of these categories has been further divided into families based on a crystalographic structure. The full list of the identified materials is provided in Table A1, Table A2, Table A3 and Table A4 in the Appendix B along with the descriptors and parameters essential for the analysis of the electronic structure phenomena behind half-metallicity. We have reported crystal lattice, saturation magnetization (M${}_{s}$), spin polarization (P${}_{0}$), energy gap (E${}_{\mathrm{gap}}$), as well as VBM and CBM of the insulating spin channel (either in spin majority or minority, as denoted by the arrows preceding the values). Finally, we have extracted quantities useful to determine the type of hybridization and strength of half-metallicity. The atom- and orbital-resolved spin magnetic moments are defined as fractions of the total magnetic moment per unit cell. We have also listed the overlaps of partial density of states functions projected on different orbitals, calculated as a percentage of a common area between each pair of functions in the metallic spin channel. These additional data are provided in the Appendix A.

### 2.1. Binary Oxides

Binary compounds are the simplest half-metallic structures; yet no element can be a half metal. However, few binary oxides beyond the representative CrO${}_{2}$ are known to exhibit half-metallicity. The high-throughput search has not greatly improved this status, as binaries account for only a small fraction of revealed materials. Moreover, among the 13 compounds listed in Table A1, several are polytypes or almost identical structures. One example is the well-studied rutile CrO${}_{2}$ (ORC) whose half-metallic properties are a consequence of the exchange splitting larger than the occupied bandwidth. The same mechanism leads to full spin polarization in CrO${}_{2}$ (TET), which can be considered a strained variant of the former; note nearly identical parameters in Table A1. Similarly, most of the Fe${}_{3}$O${}_{4}$ phases can be associated with the cubic magnetite structure (FCC) above the Verwey transition, which is ferrimagnetic due to the presence of two different ions Fe(3+) and Fe(2+); Fe${}_{3}$O${}_{4}$ is a well known half metal with the highest reported $T_{C}$ > 800 K. Last, stoichiometrically different Fe${}_{2}$O${}_{3}$ was previously found to be antiferromagnetic and insulating, thus the crystal ground state is not a half metal.

Table A1 contains several materials that have never been proposed as potential half metals. However, a detailed consideration of their properties indicates that the half-metallicity might be unfeasible in most of them. Both reported phases of CoO are antiferromagnetic [22]. Moreover, the vanadates do not manifest ferromagnetic ground states. The simpler VO${}_{2}$ is non-magnetic and shows a strong metal-to-insulator (MIT) transition at 340 K, accompanied by a structural change from tetragonal to monoclinic. The latter was previously suggested to be metallic, but we emphasize that its electronic structure is still under debate. V${}_{6}$O${}_{13}$ with its mixed-valence state V(4+) and V(5+) is again a MIT system, shown to be antiferromagnetic below 50 K and ferrimagnetic at higher temperatures; half-metallicity of the ferrimagnetic phase has never been reported. Finally, we conclude that the only binary compound that indeed seems to be half-metallic is Ce${}_{7}$O${}_{12}$, one of few rare-earth oxides revealed in this study. The crystal structure is rather complex, whereby inequivalent Ce sites are likely to cause a ferrimagnetic ordering.

### 2.2. Ternary Oxides

The diversity of the revealed ternaries is reflected in complex chemical formulas that can be found in Table A2. Although most of the structures belong to one of three main crystal families, including (i) spinels (AB${}_{2}$O${}_{4}$), (ii) perovskites (ABO${}_{3}$), and (iii) pyrochlores (A${}_{2}$B${}_{2}$O${}_{7}$), there are materials containing more than seven and up to 20 oxygen atoms in the unit cell; many of them have not been considered as candidates for half metals and can be thus regarded as new prototypes. Below, we have briefly characterized the known structural families in which we have found several new HM.

#### 2.2.1. Spinels

Spinels are cubic lattice structures characterized by a general formula AB${}_{2}$O${}_{4}$ [23,24]. Multiple degrees of freedom present in these complex crystals can be used to engineer their physical properties. In particular, they exhibit complex magnetic properties ranging from ferrimagnetism and ferromagnetism, to strong-magnetostructural coupling which is often related to the occupation of the magnetic ions in two different sublattices. According to their cation distributions, spinels can be categorized as “normal” and “inverted” spinel structures [25]. In the normal spinel structures, one-eighth of the tetrahedral interstices in oxygen sublattice are occupied by the *A* atoms and one-half of the octahedral interstices are occupied by the *B* atoms. In the inverted spinel structures, tetahedral interstices are occupied by *B* atoms and the octahedral interstices are occupied by both *A* and *B* atoms. The HT search revealed in total 18 spinels; most of them were never proposed as HM candidates.

#### 2.2.2. Perovskites

Half-metallic ternary perovskite oxides are very rare. The crystals sharing the general formula ABO${}_{3}$ contain BO${}_{6}$ octahedra whereby B cations are surrounded by oxygen atoms. Such a configuration causes a superexchange mechanism mediated by dominating oxygen atoms, which makes the magnetic state more likely to be antiferromagnetic than ferromagnetic. Thus, the electronic structure is usually semiconducting and insulating. It has been though proven that the half-metallic state could emerge upon doping, strain, or intrinsic defects. The most known example is a non-stoichiometric La${}_{x}$Sr${}_{1-x}$MnO${}_{3}$, based on antiferromagnetic and insulating LaMnO${}_{3}$ in which the sufficient Sr doping may yield half-metallicity [15]. The HT search revealed 23 potentially half-metallic perovskites that could be interesting for similar non-stoichiometric-doped configurations. Indeed, we have identified BaFeO${}_{3}$ recently reported to be ferromagnetic and half-metallic below $T_{C}$∼180 K [26]. We have also listed BaRuO${}_{3}$, which was shown to be ferromagnetic [27], while the previous prediction of half-metallicity still awaits experimental verification [28]. Finally, we have revealed stoichiometric SrRuO${}_{3}$, which was theoretically predicted to be half-metallic under doping with Sn or Ti [29,30].

#### 2.2.3. Pyrochlores

The most representative example of a half-metallic pyrochlore is Tl${}_{2}$Mn${}_{2}$O${}_{7}$ [16]. However, it has not been listed in Table A2, as the electronic structure from the aflowlib database does not comply with the criteria for robust half-metallicity adapted in the HT search. In particular, the stringent condition imposed on the density of states eliminates the materials that would have more than 0.005 states/eV at energies within the band gap of the insulating channel. The closer inspection of the calculated electronic structure confirms that Tl${}_{2}$Mn${}_{2}$O${}_{7}$ indeed does not fit in this regime; this compound could only be included upon increasing the threshold. Nevertheless, we have found 11 different pyrochlores with the half-metallic electronic structure.

### 2.3. Quaternary Oxides

The majority of the quaternary oxides reported in Table A3 belongs to the crystal family group of the double perovskites with the general formula A${}_{2}$BB’O${}_{6}$, consisting of two different perovskites—ABO${}_{3}$ and AB’O${}_{3}$—arranged in a three-dimensional checkerboard pattern. The possibility of choosing two different transition metal ions opens up a wide range of possibilities to tailor the magnetic properties in this class of materials. The most known compound from this family is Sr${}_{2}$MoFeO${}_{6}$ with a large magnetic moment and Curie temperature of more than 420 K [17,31]. The HT search revealed over 30 double perovskites, including a similar Sr${}_{2}$MoCoO${}_{6}$ structure. The mechanism that determines half-metallicity in these compounds is quite complex and related to a combination of superexchange interaction in the B-O-B’ chains and the hybridization of transition metal orbitals with O *2p* states. The magnetic properties of double perovskites are, however, quite well known [32]. In addition, Table A1 contains previously unrecognized prototypes, many of which seem to be good candidates for half metals.

### 2.4. Quinary Oxides

Finally, we note that the search revealed only four quinary compounds. Such a result does not mean that quinary half-metallic oxides would not occur in nature. The reason is mostly related to a currently limited number of quinary materials in the aflowlib repository. In particular, we have noticed that a known half-metallic quadruple perovskite CaCu${}_{3}$Fe${}_{2}$Re${}_{2}$O${}_{12}$ has not been found because it is yet absent in the database [33]. The materials listed in Table A4 are rather difficult to analyze without performing additonal calculations. Although the radioactive CH${}_{2}$P${}_{2}$PuO${}_{6}$ is useless for spintronics, CuH${}_{12}$Mn${}_{2}$N${}_{4}$O${}_{8}$, H${}_{4}$K${}_{2}$N${}_{4}$PdO${}_{10}$, and La${}_{3}$MnS${}_{3}$WO${}_{6}$ indeed seem to be robust half metals. The latter can be considered a quasi-1D spin chain and would probably reveal short range magnetic ordering below 4 K. The ground states still need to be verified but these structures are clearly new prototypes of half metals (see Figure 2).

## 3. Discussion

The presented search based on the HT screening suggests that several new half-metallic oxides could be discovered. In fact, the choice of oxides as target materials for the above analysis was deliberate in a larger perspective of rapidly evolving oxide spintronics [34]. TMO often host a large variety of physical phenomena that emerge due to the complex interplay of the electronic charge, spin, and orbital degrees of freedom. Beyond magnetic properties, more exotic effects have been extensively studied, including multiferroicity, superconductivity, or magnetocaloric behavior [35,36,37]. Thus, the role of half-metallic oxides does not need to be limited solely to the generation of spin-polarized currents; being highly multifunctional, they could be capable to perform multiple tasks within one spin-based device [38]. The interplay of diverse phenomena in realistic half-metallic systems is therefore an interesting direction to explore in more specific studies.

Importantly, realization of either novel or conventional spintronics devices operating at room temperature requires robust half-metallicity. One could therefore raise a question, how many of the identified compounds will be still half-metallic at 300 K? Our search and analysis based primarily on the electronic structure information from the aflowlib cannot give a precise answer at the present stage. As we have previously explained, most of the magnetic compounds in materials databases are in a ferromagnetic configuration, whereas, in reality, oxides often exhibit antiferromagnetic or more complex magnetic ground states. Although a large number of the identified materials may indeed be ferromagnetic and several are confirmed half metals, a complete verification of the magnetic phase along with the transition temperature would be desirable [39,40,41]. Multi-step calculations of numerous hypothetical magnetic configurations for each compound are though a great challenge. In a short-term perspective, exploring particular oxide-based interfaces could be more appropriate than the verification of the whole dataset ground states.

## 4. Methods

The high-throughput screening has been performed utilizing the aflux search engine, which helps to extract the electronic structure data from the aflowlib database [42]. In particular, we have analyzed the output files of density functional calculations performed within the aflow framework [43,44,45], which leverages the Vienna Ab initio Simulation Package (VASP) [46,47]. Projector-augmented-wave pseudopotentials were used to treat electron–ion interactions [48]; kinetic energy cut-offs were set to highest recommended value among corresponding pseudopotentials. The exchange–correlation interaction was treated in the generalized gradient approximation in the parametrization of Perdew, Burke, and Ernzerhof (PBE) [49]. LSDA+U approach in formulation of Dudarev [50] was used to account for electronic correlations of transition metal ions (U parameters are listed in [19]). Spin–orbit interaction was not included in the calculations.

The search procedure has been described in Results; we hereby give technical details which determine the exact output of each phase, necessary to reproduce the list of compounds extracted from aflowlib. In the first step, based on the condition P${}_{0}$(E${}_{F}$) = ≠ 0, we select thousands of magnetic materials with spin imbalance at the Fermi level. Certainly, such a descriptor cannot provide sufficient information about the half-metallic state. Several materials that satisfy this criterion may not have a robust band gap. For instance, binary compound CuO (ICSD-628616) switches the conductance between two spin channels just around $E_{F}$, and reveals semi-metallic rather than half-metallic properties. The essential phase of screening relies on the analysis of DOS. Materials with more (less) than 0.005 states/eV at energies within the band gap in the insulating (conducting) channel around $E_{F}$ are screened out as not half-metallic (note that this criterion eliminates a known half metal Tl${}_{2}$Mn${}_{2}$O${}_{7}$, see Section 2.2.3). As an outcome of this phase, we have revealed 1061 compounds, among which 494 are oxides. Finally, we have selected materials referred to as strong half metals potentially useful for spintronics, whose spin imbalance is sufficiently large within the energy window of the band gap and robust against thermal fluctuations. Screening based on the set of constraints (P${}_{0}$(E${}_{\mathrm{gap}}$) > 0.8; E${}_{g}$ < 3.5 eV; −0.01 > VBM > −1.5 eV; 0.1 < CBM < 2.5 eV) revealed 223 oxides reported in Table A1, Table A2, Table A3 and Table A4.

## 5. Conclusions

In summary, we have explored the aflowlib repository containing electronic structure data of thousands of materials, and identified over one hundred potentially half-metallic oxides. A large number of these compounds were not previously recognized as half metals. Remarkably, we have revealed new crystal prototypes, suggesting a way to design additional half-metallic oxides sharing the same structure. Finally, we have also indicated numerous crystals belonging to the same families as some of the known half-metallic compounds. These include newly identified spinels, perovskites, pyrochlores, and double perovskites. The quantitative analysis of the electronic structure has indicated a strong *p-d* hybridization as a common mechanism behind half-metallicity in a vast majority of the considered compounds. We believe that this study will stimulate further exploration of a larger chemical space as well as an ultimate confirmation of half-metallicity in selected structures.

## Figures and Tables

**Figure 1 molecules-25-02010-f001:**
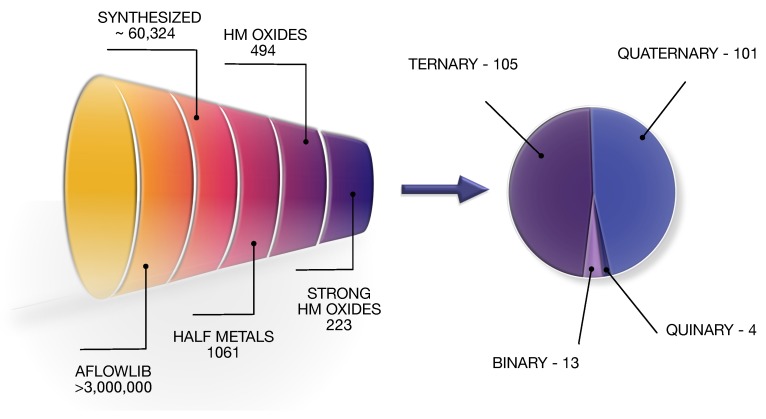
Left: Scheme illustrating steps of the high-throughput (HT) search for half-metallic oxides starting with the aflowlib database. Right: Diagram representing the distribution of different structural groups among the identified half-metal candidates.

**Figure 2 molecules-25-02010-f002:**
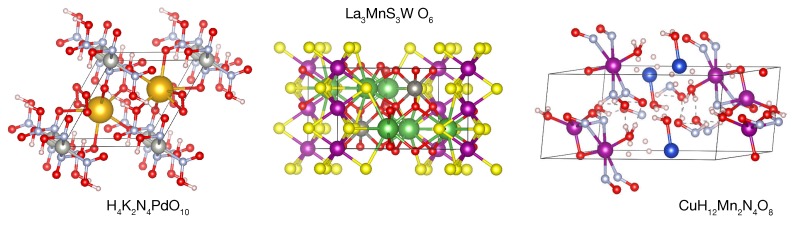
Crystal structures of quinary prototypes of half metals revealed in the HT search.

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
