# Peer review of "High-Throughput Computational Search for Half-Metallic Oxides"

_molecules, 2020, doi:10.3390/molecules25092010_

Round 1

Reviewer 1 Report

The paper brings interesting new results regarding the half-metallic

oxides. The authors perform a high-throughput computational search

for half-metallic compounds and identify new potential half metals.

The paper is written concisely and comprehensively and the results

are sufficiently novel and of broad interest to warrant the publication

in Molecules. I recommend its publication in the present form.

Author Response

We are glad that the Reviewer recognized the quality of the submitted work.

Reviewer 2 Report

The authors present the results of a high-throughput computational search of half metallic oxydes, a class of materials with interesting electric and magnetic properties that are ideal candidates for spintronic applications. 

Their search defines a number of parameters that can be used to identify half-metals, that are searched for in the AFLOWLIB database.

The paper is concise, well written and clear. The results are presented in an adequate, orderly and informative way and are most certainly of interest for the specialized community. 

I support the publication of this paper in its current form.

Author Response

We thank the Reviewer for recommending publication of the manuscript in its present form.

Reviewer 3 Report

The manuscript presents and discusses the throughput of a computational search for half-metallic oxides. The text is clearly prepared and can be followed with no issues

The content and objective of the work is of high-quality and deserves its publication. My only recommendation is some expansion on section 4 (methods) providing maybe two examples of cases that resulting in their inclusion in the list presented in the appendix or its exclusion (although explained the work guidelines, not all audience will be familiar with the whole process so that a chart can enhance the readability of the manuscript).

Author Response

We are grateful for the positive assessment of the paper. Following the suggestion of the Reviewer, we have added a new paragraph in Section IV which provides further technical details of the screening as well as examples of inclusion/exclusion useful to guide a less familiar reader. The added text is given in lines 215-229 on page 6 of the revised manuscript.